# Potentially avoidable causes of hospitalisation in people with dementia: contemporaneous associations by stage of dementia in a South London clinical cohort

Usha Gungabissoon ![ORCID] ,[1,2] Gayan Perera ![ORCID] ,[2] Nicholas W Galwey,[3] Robert Stewart[2,4]

[1]Epidemiology (Value Evidence and Outcomes), GSK, Brentford, UK
[2]Psychological Medicine, King's College London, Institute of Psychiatry Psychology and Neuroscience, London, UK
[3]Research Statistics, GSK, London, UK
[4]Mental Health of Older Adults, South London and Maudsley NHS Foundation Trust, London, UK

**Correspondence to**
Usha Gungabissoon;
usha.2.gungabissoon@gsk.com

## ABSTRACT

**Objectives** To estimate the frequency of all-cause and ambulatory care sensitive condition (ACSCs)-related hospitalisations among individuals with dementia. In addition, to investigate differences by stage of dementia based on recorded cognitive function.

**Setting** Data from a large London dementia care clinical case register, linked to a national hospitalisation database.

**Participants** Individuals aged ≥65 years with a confirmed dementia diagnosis with recorded cognitive function.

**Outcome measures** Acute general hospital admissions were evaluated within 6 months of a randomly selected cognitive function score in patients with a clinical diagnosis of dementia. To evaluate associations between ACSC-related hospital admissions (overall and individual ACSCs) and stage of dementia, an ordinal regression was performed, modelling stage of dementia as the dependant variable (to facilitate efficient model selection, with no implication concerning the direction of causality).

**Results** Of the 5294 people with dementia, 2993 (56.5%) had at least one hospitalisation during a 12-month period of evaluation, and 1192 (22.5%) had an ACSC-related admission. Proportions with an all-cause or ACSC-related hospitalisation were greater in the groups with more advanced dementia (all-cause 53.9%, 57.1% and 60.9%, p 0.002; ACSC-related 19.5%, 24.0% and 25.3%, p<0.0001 in the mild, moderate and severe groups, respectively). An ACSC-related admission was associated with 1.3-fold (95% CI 1.1 to 1.5) increased odds of more severe dementia after adjusting for demographic factors. Concerning admissions for individual ACSCs, the most common ACSC was urinary tract infection /pyelonephritis (9.8% of hospitalised patients) followed by pneumonia (7.1%); in an adjusted model, these were each associated with 1.4-fold increased odds of more severe dementia (95% CI 1.2 to 1.7 and 1.1 to 1.7, respectively).

**Conclusions** Potentially avoidable hospitalisations were common in people with dementia, particularly in those with greater cognitive impairment. Our results call for greater attention to the extent of cognitive status impairment, and not just dementia diagnosis, when evaluating measures to reduce the risk of potentially avoidable hospitalisations.

## Strengths and limitations of this study

► Through linkage of a catchment specialist healthcare case register to a national hospitalisation database, our study benefited from near-complete outcome ascertainment enabling us to evaluate potentially avoidable causes of hospitalisation in a large sample of clinically confirmed cases of dementia.

► Severity of dementia was ascertained from routinely administered cognitive assessments, making use of natural language processing algorithms with established validity.

► Reasons for hospitalisation were derived from the discharge diagnoses recorded for last episode of the hospital admission, meaning it was not possible to distinguish complications arising during the hospital stay from reasons for the initial admission.

► Data were obtained from single service provider in an urban catchment area, which may limit the generalisability of our findings to other settings.

► The analyses did not account for comorbidities, use of medications or living arrangement (such as home vs care home); furthermore, dementia stage and hospitalisations were measured contemporaneously, meaning that conclusions about causation should not be drawn.

## INTRODUCTION

Dementia represents a significant challenge for health services. For example, in the UK National Health Service (NHS), an estimated 25% of acute beds are occupied by people with dementia,[1] and recent analyses have shown a 35% increase in the number of emergency hospital admissions for people with dementia in England over the past 5 years.[2] In 2016, the global economic costs caused by dementia approached US $1 trillion, with the highest economic burden seen in Europe and North America.[3] Major contributing factors to costs include

disease severity and higher rates of hospitalisation in people with dementia relative to older people without dementia.[4]

While some hospitalisations are necessary and expected, others are potentially avoidable. Ambulatory care sensitive conditions (ACSCs) are conceptualised as chronic and acute health conditions such as heart failure, pneumonia, pyelonephritis and complications of diabetes, for which effective primary and community care may avoid the need for hospital admission by preventing the onset of these diseases, controlling the acute episodic illness or managing the chronic condition more effectively.[5 6] ACSC-related admissions, for example, are used to assess how the NHS is performing in England.[7]

Preventing hospitalisation is of particular importance in people with dementia; hospitalisation is associated with poorer outcomes, including prolonged hospital stays and increased risk of complications relative to people without dementia.[8–11] Furthermore, hospitalisation itself can trigger distress, or confusion which can contribute to a decline in functioning creating new care challenges or can intensify the existing burden of care.[12]

Even at early stages of cognitive impairment, a person's ability to make healthcare decisions can be altered,[13] including changes in health-seeking behaviours and management of existing conditions.[14] This could explain why the age-adjusted risk of potentially avoidable hospitalisations is greater in people with dementia than those without.[9 10] There is also some evidence suggesting that the risk of potentially avoidable hospitalisations is greater in people with later-stage dementia.[8 15] However, previous studies have been small, used non-representative convenience samples or relatively crude methods to evaluate cognitive deficits. Understanding which potentially avoidable causes of hospitalisation predominate at later stages of dementia can help to tailor healthcare needs in this population in order to prevent the need for hospitalisation.

In the present study, we used data drawn from a mental healthcare case register linked to a national hospitalisation episodes database, to describe ACSC-related hospitalisations, a proxy for potentially avoidable hospitalisations, among a large sample of patients with dementia. Specifically, we first sought to investigate whether ACSC-related hospital admission was associated with more advanced stages of dementia, and second to explore whether hospitalisations for specific ACSCs were associated with more advanced dementia stage.

## METHODS
### Study design
This study investigated acute general hospital admissions around the time of a given cognitive function score (index date) in a sample of patients with a clinical diagnosis of dementia using linked clinical datasets.

### Data sources and sample
The study sample comprised of residents within the South London and Maudsley NHS Foundation Trust (SLaM) who had received a dementia diagnosis in specialist services. The trust is a provider of mental healthcare, dementia assessment and management, for a south London catchment of >1.2 million residents in Lambeth, Lewisham, Croydon and Southwark; electronic health records (EHRs) have been implemented across all SLaM services since 2006.[16] The Clinical Record Interactive Search (CRIS) data resource, used to identify dementia cases, provides research access to anonymised EHRs ; it allows both structured and unstructured data to be abstracted from patient records based on interactions within secondary care mental health services within SLaM.

Routine diagnoses are recorded using the International Classification of Diseases 10th edition (ICD-10) coding system and are supplemented in CRIS by a natural language processing (NLP) to abstract measures of cognitive impairment.[16 17] Hospital outcomes (hospital admissions and their causes) for patients with dementia were obtained from the a linkage with the Hospital Episode Statistics (HES) database. HES contains details of all inpatient admissions at NHS hospitals in England.[18] Discharge diagnoses are also recorded as ICD-10 codes and are available for each hospitalisation episode. HES data up to 31 March 2013 were available at the time of analysis.

### Study population
Patients with a diagnosis of dementia first recorded in SLaM between 1 January 2006 and 30 September 2012, and aged ≥65 years at diagnosis, were identified on CRIS. Dementia was defined on the presence of F00*, F01*, F02*or F03* ICD diagnosis codes recorded up to the sixth position in the structured data. Eligible patients were those who had at least one cognitive assessment (from structured data or free text using NLP) recorded: either a Mini Mental State Examination score (MMSE) or a cognitive impairment score from the Health of the Nation Outcome Scales (HoNOS). HoNOS is a clinician-rated instrument usually completed on a structured field within the EHR at first assessment, which contains subscales rated 0 (no problem) to 4 (severe/very severe problem); its cognitive scale correlates with MMSE measurement.[19] To exclude patients with potential mild cognitive impairment rather than dementia, the sample was further restricted to those with an MMSE score of <28 or HoNOS cognitive impairment score >1. In addition, we excluded patients who received their first dementia diagnosis from an acute hospital liaison service; these reflected cases where the recording of a dementia diagnosis might have been precipitated by a hospitalisation and by definition be a group with significant hospitalised physical disorders, therefore, potentially biasing any associations between dementia and comorbidity.

For each eligible patient, we randomly selected a single measure of cognition recorded in the patient's clinical

record using a random number generator in Microsoft Excel; the date for the measure was assigned as the index date. This process was applied hierarchically: if several MMSE scores were recorded over time for a given patient, then one was randomly selected; for those individuals without an MMSE measurement, a random HoNOS cognitive impairment subscale was selected in the same manner. The randomly selected score for each patient was used to categorise the stage of dementia into mild, moderate or severe on the date of the cognitive assessment as described below.

## Measurements

The following demographic information was obtained from CRIS: age at diagnosis of dementia, sex, ethnicity (White-European, other, not specified), dementia diagnostic group (Alzheimer's, vascular, mixed or other) and Index of Multiple Deprivation (IMD) score, a widely used measure of relative neighbourhood deprivation.[20] IMD is derived from Census data and calculated for clusters of adjacent unit postcodes in England known as Lower layer Super Output Areas, standard geographic units with around 1500 residents; a higher IMD score indicates a more deprived area.

To classify dementia stage at the index date, we used cut-offs similar to those used by others,[21] defining MMSE scores of >21–27, 10–20 and <10, or a HoNOS cognitive impairment subscale score of 2, 3 and 4, as representing mild, moderate or severe dementia, respectively.

Acute general hospital inpatient admissions were obtained for the 6-month period either side of the index date, that is, over a 12-month period. Hospitalisations were defined from HES episodes, combining contiguous episodes (ie, where start and end dates were on the same day); in the context of this analysis, an admission, or spell, is defined as a continuous period spent as a patient in hospital. Four-digit ICD-10 discharge diagnoses recorded during the last episode of each hospital admission were obtained (up to 20 diagnoses may be recorded as part of a hospital episode).

ACSCs were identified based on the presence of specific ICD-10 codes and their position in the record, using a modification of the approach used by Bardsley *et al* and the UK Health and Social Care Information Centre; for example, angina is based on the presence of specific ICD-10 codes in the primary position, whereas pneumonia is defined based on the presence of a relevant code in any position.[5 6] Deviations from the published approach comprised the following: (1) for the urinary tract infection (UTI) category, we deemed cystitis to be relevant and, therefore, included relevant codes in this group, (2) we removed O15 (eclampsia) from the convulsions and epilepsy category as used by Bardsley. Each ACSC was specified individually as a binary variable to flag the presence of a hospitalisation with that specific condition. Given that diagnoses may not be recorded in the order of clinical importance in the real-world setting, as a sensitivity analysis, we relaxed the requirement for conditions which were based on a diagnosis in the primary position only, to include diagnoses in either of the first two positions. Online supplemental table 1 lists the ACSCs, the ICD codes and the position of the ICD code.

Additional covariates included binary variables to indicate an all-cause or any ACSC-related hospitalisation during the 12-month observation period. Time since first recorded dementia diagnosis (a proxy for disease duration) in SLaM at index date was also obtained.

## Statistical analyses

Microsoft Excel was used for data management, and Stata V.15·0 was used for statistical analyses. Patient characteristics were described by stage of dementia; normally distributed variables were presented as mean±SD and non-normally distributed variables as median and upper and lower quartiles. For the bivariate analysis, comparison of continuous variables between dementia stages (mild, moderate, severe) was evaluated using analysis of variance (ANOVA), with the exception of number of inpatient admissions and time since presentation at SLaM, which were evaluated by the Kruskal-Wallis test due to their non-normal distribution. For the relationship between each categorical variable and stage of dementia, a $\chi^2$ test of association was performed, and percentage distributions were presented.

To evaluate the association between any ACSC-related hospitalisation and contemporaneously measured stage of dementia, unadjusted, age–sex-adjusted and fully adjusted OR were obtained from generalised ordinal logistic regression models,[22] with dementia stage (mild, moderate, severe) as the dependant variable. Including dementia stage as the outcome variable was deemed to be the most efficient approach to evaluate ACSC-related hospitalisations and specific causes of ACSC-related admissions as it allowed us to use the efficient, flexible approach of stepwise model selection, to evaluate the importance of the relationship with other variables. Because our aim was to assess cotemporal associations between stage and hospitalisations rather than causality, we considered that whether dementia stage was modelled as a response, or as an explanatory variable, was not critical.

To investigate whether specific causes of ACSC-related admissions were associated with dementia stage, a backward stepwise model selection was used to build a multivariable generalised ordinal logistic regression model, with dementia stage as the dependant variable. We used the gologit2 STATA command with the autofit option to selectively relax the model, removing the proportional odds assumption for any variable for which this assumption was violated.[22] Sex, time from first dementia record at index date (years) and age-group at index date were deemed clinically relevant and were, therefore, forced into the model (ie, retained regardless of their p value). Variables

with p<0.1 in the bivariate analysis were initially included in the stepwise model selection. During stepwise selection, variables with p<0.2 were retained in the model, and following exclusion of any variable, the variables already excluded were considered for reinclusion. Collinearity was evaluated based on the variance inflation factor (VIF) and tolerance for each variable; covariates with a VIF greater than 2.5 were removed one at a time, starting with variables with the largest VIF, and the impact on remaining covariates was assessed until no remaining variable had a VIF >2.5. Furthermore, interactions between age-group and sex, and between age-group and dementia diagnostic group, were assessed for possible inclusion, based on the Akaike information criterion (AIC).

## Patient and public involvement

Patients or the public were not involved in the design, or conduct, or reporting or dissemination plans of our research.

## RESULTS

In total, 5294 patients with dementia met the study criteria and were included in the analysis sample. Almost two-thirds of the sample were women (N=3393, 64.1%) and the mean age at the date of randomly chosen cognitive assessment was 82.5 years (SD 6.8). The proportions of patients with mild, moderate and severe dementia at these index dates were 37.3%, 46.4% and 16.1%, respectively. Descriptive characteristics by stage of dementia are presented in table 1.

## All-cause and ACSC-related hospitalisations

Overall, 56.5% (N=2993 of 5,294) of individuals analysed had at least one inpatient admission (of any cause) in the 6-month period either side of the index date, and this percentage was greater with more advanced dementia: 53.9% of patients with mild, 57.1% with moderate and 60.9% with severe dementia ($\chi^2$ (df 2)=12.47, p=0.002) (table 2).

Within the cohort, 22.5% had at least one ACSC-related hospitalisation (N=1192/5,294), which accounted for almost 40% of patients with any hospital admission (N=1192/2,993). ACSC-related hospitalisations were observed in 36.2%, 42.0% and 41.5% of hospitalised patients with mild, moderate and severe dementia, respectively ($\chi^2$ (df 2)=9.25 p=0.01). The most common ACSC was UTI/pyelonephritis (9.8% of hospitalised patients) followed by pneumonia (7.1%), both of which tended to be more frequent with more severe dementia (p<0.0001 for both conditions). The mean number of ACSC admissions among hospitalised patients was highest in those with moderate (0.67, SD 1.04) compared with mild or severe dementia (0.57, SD 1.01 and 0.58, SD 0.85, F (2, 2990)=3.33, p=0.036).

| Table 1 | Descriptive analysis by dementia severity (N=5294) | | | | | |
|---|---|---|---|---|---|---|
| | | **Mild (N=1979)** | **Moderate (N=2461)** | **Severe (N=854)** | **All (N=5294)** | **Test statistic (d.f) p value** |
| Age at index date N (%) | 65-74 years | 326 (16.5) | 292 (11.9) | 116 (13.6) | 734 (13.9) | $\chi^2$(4)= 23.35 p<0.0001 |
| | 75-84 years | 892 (45.1) | 1107 (45.0) | 397 (46.5) | 2396 (45.3) | |
| | 85+ | 761 (38.5) | 1062 (43.2) | 341 (39.9) | 2164 (40.9) | |
| Mean age at index date (SD) | | 81.9 (6.9) | 82.9 (6.8) | 82.4 (6.9) | 82.5 (6.8) | F(2, 5291)= 11.96 p<0.0001 |
| Sex | Female | 1233 (62.3) | 1615 (65.6) | 545 (63.8) | 3393 (64.1) | $\chi^2$(2)=5.29 p 0.071 |
| | Male | 746 (37.7) | 846 (34.4) | 309 (36.2) | 1901 (35.9) | |
| Ethnic group | White-European | 1614 (81.6) | 1915 (77.8) | 655 (76.7) | 4184 (79.0) | $\chi^2$(4)= 9.54 p 0.008 |
| | Other | 337 (17.0) | 502 (20.4) | 172 (20.1) | 1011 (19.1) | |
| | Not specified | 28 (1.4) | 44 (1.8) | 27 (3.2) | 99 (1.9) | |
| Deprivation score—mean (SD) | | 26.02 (11.4) | 27.46 (11.0) | 28.10 (11.0) | 27.02 (11.2) | F(2, 5260)= 13.75 p<0.0001 |
| Time (years) since first presentation at SLaM—median (IQR) | | 0.31 (0.03, 0.91) | 0.31 (0.02, 1.21) | 0.31 (0.02, 1.28) | 0.31 (0.02, 1.06) | H (2)=1.04 p 0.596 |
| Dementia diagnostic group | Alzheimer's | 852 (43.1) | 931 (37.8) | 336 (39.34) | 2119 (40.03) | $\chi^2$(6)= 25.81 p<0.0001 |
| | Mixed | 206 (10.4) | 278 (11.3) | 74 (8.7) | 558 (10.5) | |
| | Vascular | 385 (19.5) | 602 (24.5) | 197 (23.1) | 1184 (22.4) | |
| | Other | 536 (27.1) | 650 (26.4) | 247 (28.9) | 1433 (27.1) | . |

Percentages presented in parentheses unless otherwise specified.
Test statistics: $\chi^2$ test of association, F: one-way ANOVA, H: Kruskal-Wallis test.
ANOVA, analysis of variance; d.f, degrees of freedom; SLaM, South London and Maudsley NHS Foundation Trust.

**Table 2** Hospital admissions by severity of dementia (N=5294)

| | Mild (N=1979) | Moderate (N=2461) | Severe (N=854) | All (N=5294) | Test statistic (d.f) p value |
|---|---|---|---|---|---|
| Inpatient admission±6 months of index date, N (%) | 1067 (53.9) | 1406 (57.1) | 520 (60.9) | 2993 (56.5) | $\chi^2(2)$= 12.47, p 0.002 |
| Number of all cause admissions per patient—median (lower and upper quartile)* | 2 (1,3) | 2 (1,3) | 2 (1,3) | 2 (1,3) | H(2)= 3.42, p 0.181 |
| Ambulatory care sensitive conditions† N (% of all patients) | | | | | |
| Angina | 27 (1.4) | 28 (1.1) | 5 (0.6) | 60 (1.1) | $\chi^2(2)$= 3.2, p 0.199 |
| Asthma | 2 (0.1) | 2 (0.1) | 1 (0.1) | 5 (0.1) | $\chi^2(2)$= 0.10, p 0.951 |
| Cellulitis | 19 (1.0) | 27 (1.1) | 9 (1.1) | 55 (1.0) | $\chi^2(2)$= 0.20, p 0.904 |
| Congestive heart failure | 28 (1.4) | 38 (1.5) | 10 (1.2) | 76 (1.4) | $\chi^2(2)$= 0.63, p 0.729 |
| Convulsion/epilepsy | 16 (0.8) | 23 (0.9) | 12 (1.4) | 51 (1.0) | $\chi^2(2)$= 2.26, p 0.322 |
| COPD | 28 (1.4) | 40 (1.6) | 8 (0.9) | 76 (1.4) | $\chi^2(2)$= 2.13, p 0.344 |
| Dehydration | 8 (0.4) | 20 (0.8) | 7 (0.8) | 35 (0.7) | $\chi^2(2)$= 3.18, p 0.204 |
| Dental | 9 (0.5) | 10 (0.4) | 1 (0.1) | 20 (0.4) | $\chi^2(2)$= 1.91, p 0.385 |
| Diabetes complications | 30 (1.5) | 48 (2.0) | 11 (1.3) | 89 (1.7) | $\chi^2(2)$= 2.21, p 0.332 |
| ENT infections | 5 (0.3) | 2 (0.1) | 1 (0.1) | 8 (0.2) | $\chi^2(2)$= 2.21, p 0.331 |
| Gastroenteritis | 21 (1.1) | 20 (0.8) | 3 (0.4) | 44 (0.8) | $\chi^2(2)$= 3.67, p 0.16 |
| Hypertension | 3 (0.2) | 3 (0.1) | – | 6 (0.1) | $\chi^2(2)$= 1.24, p 0.538 |
| Iron deficiency anaemia | 6 (0.3) | 15 (0.6) | 2 (0.2) | 23 (0.4) | $\chi^2(2)$= 3.32, p 0.19 |
| Other vaccine preventable | 2 (0.1) | – | – | 2 (0.04) | $\chi^2(2)$= 3.35, p 0.187 |
| Perforated/bleeding ulcer | – | 3 (0.1) | 3 (0.4) | 6 (0.1) | $\chi^2(2)$= 6.53, p 0.038 |
| Pneumonia | 103 (5.2) | 201 (8.2) | 70 (8.2) | 374 (7.1) | $\chi^2(2)$= 16.65, p<0.0001 |
| UTI/pyelonephritis | 155 (7.8) | 255 (10.4) | 111 (13.0) | 521 (9.8) | $\chi^2(2)$= 19.34, p<0.0001 |
| Any ACSC admission N (%) | | | | | |
| Overall | 386 (19.5) | 590 (24.0) | 216 (25.2) | 1192 (22.5) | $\chi^2(2)$= 17.06, p<0.0001 |
| Among those with an inpatient admission | 386 (36.2) | 590 (42.0) | 216 (41.5) | 1192(39.8) | $\chi^2(2)$= 9.25, p 0.010 |
| Mean number of ACSC admissions per patient (SD) | | | | | |
| Among those with an inpatient admission | 0.6 (1.0) | 0.7 (1.0) | 0.6 (0.9) | 0.6 (1.0) | F(2, 2990)= 3.33 p 0.036 |

Test statistics: $\chi^2$ test, F: one-way ANOVA, H: Kruskal-Wallis test.
*Among hospitalised.
†No admissions meeting the criteria for influenza or gangrene identified.
ACSC, ambulatory care sensitive condition; ANOVA, analysis of variance; COPD, chronic obstructive pulmonary disease; d.f, degrees of freedom; ENT, ear nose and throat; UTI, urinary tract infection.

## Associations between any ACSC-related hospitalisation and dementia stage

Any ACSC-related hospitalisation was significantly associated with more advanced dementia, that is, an OR significantly greater than 1 for moderate/severe versus mild or severe versus mild/moderate in the unadjusted model (unadjusted OR 1.28, 95% CI 1.14 to 1.45). The association remained significant in the fully adjusted model, which accounted for age, sex, time since first presentation at SLaM, ethnic group and neighbourhood deprivation (OR 1.27, 95% CI 1.12 to 1.44 for both moderate/severe vs mild and severe vs mild/moderate) (table 3).

## Exploratory analysis: associations between specific ACSC-related hospitalisations and dementia stage

Variables included in the multivariable model after stepwise selection are summarised in table 4. The goodness of fit of the final model, as measured by the likelihood ratio $\chi^2$ test, demonstrated that the model as a whole fitted significantly better than the null model (ie, a model with no predictors; likelihood ratio $\chi^2$ 144.6, p<0.0001). The output from the stepwise-selected regression model is presented in table 4. The proportional odds assumption was violated for age-group, and dementia diagnostic group, meaning that the effect of these explanatory

**Table 3** Unadjusted and adjusted ORs and 95% CIs for any ACSC-related admission by stage of dementia

| | Severe/moderate vs mild or severe vs mild/moderate dementia*, OR (95% CI) |
|---|---|
| Unadjusted | 1.28 (1.14 to 1.45) |
| Age, sex adjusted | 1.27 (1.13 to 1.44) |
| Fully adjusted† | 1.27 (1.12 to 1.44) |

*From a generalised ordinal logistic regression model: the same OR applies to both comparisons as the proportional odds assumption is not violated.
†Age-group, sex, time since first presentation at SLaM, ethnicity and IMD.
ACSC, ambulatory care sensitive condition; IMD, Index of Multiple Deprivation; SLaM, South London and Maudsley NHS Foundation Trust.

variables was not the same at the two dementia-severity boundaries.

The model indicated that conditional on other explanatory variables, a greater time since first presentation at SLaM, higher IMD score (ie, residence in a more deprived area) and 'other' ethnicity, were associated with greater odds of more severe dementia. Greater age was associated with greater odds of moderate/severe dementia relative to mild but not of severe dementia relative to mild/moderate.

Considering specific conditions, perforated/bleeding ulcer, UTI/pyelonephritis and pneumonia hospitalisations were all significantly associated with greater odds of more advanced dementia, that is, moderate/severe relative to mild or severe relative to mild/moderate. Although a strong association between perforated/bleeding ulcer and dementia stage (OR 7.01 p 0.01) was observed, it is important to note that in terms of absolute frequency, this condition was uncommon, being recorded in only six individuals. UTI/pyelonephritis and pneumonia were also associated with higher odds of more severe dementia (OR 1.38 p<0.0001, OR 1.38 p 0.002, respectively); this was consistent across dementia severity categories, that is, the ORs and p values apply to both cutpoints. Gastroenteritis-related admissions had an inverse relationship with dementia stage, namely, that admission was associated with a lower odds of advanced dementia stages: however, this was of borderline significance (OR 0.57 p 0.06).

**Table 4** Results from stepwise-selected partial generalised ordinal logistic regression model for severity of dementia (N=5294)*

| Explanatory variable | Moderate/severe vs mild dementia | | Severe vs mild/moderate dementia | |
|---|---|---|---|---|
| | OR (95% CI) | P value | OR (95% CI) | P value |
| Sex (male reference) | 1.07 (0.96 to 1.2) | 0.203 | 1.07 (0.96 to 1.2) | 0.203 |
| Time since first presentation at SLaM | 1.13 (1.08 to 1.19) | <0.0001 | 1.13 (1.08 to 1.19) | <0.0001 |
| Age group at index date (65–74 years reference) | | | | |
| | 1.36 (1.15 to 1.62) | <0.0001 | 1.04 (0.82 to 1.31) | 0.763 |
| | 1.53 (1.28 to 1.83) | <0.0001 | 1.01 (0.79 to 1.28) | 0.949 |
| Deprivation score | 1.01 (1.01 to 1.02) | <0.0001 | 1.01 (1.01 to 1.02) | <0.0001 |
| Dementia diagnostic group (AD reference) | | | | |
| Vascular | 1.34 (1.15 to 1.56) | <0.0001 | 1.02 (0.84 to 1.24) | 0.825 |
| Mixed | 1.16 (0.95 to 1.4) | 0.145 | 0.8 (0.61 to 1.05) | 0.114 |
| Other | 1.12 (0.98 to 1.27) | 0.099 | 1.12 (0.98 to 1.27) | 0.099 |
| Ethnic group (European reference) | | | | |
| Other | 1.2 (1.05 to 1.37) | 0.009 | 1.2 (1.05 to 1.37) | 0.009 |
| Pneumonia | 1.38 (1.13 to 1.7) | 0.002 | 1.38 (1.13 to 1.7) | 0.002 |
| Perforated/bleeding ulcer | 7.01 (1.56 to 31.57) | 0.011 | 7.01 (1.56 to 31.57) | 0.011 |
| Dehydration | 1.71 (0.92 to 3.17) | 0.09 | 1.71 (0.92 to 3.17) | 0.09 |
| Gastroenteritis | 0.57 (0.32 to 1.02) | 0.059 | 0.57 (0.32 to 1.02) | 0.059 |
| UTI/pyelonephritis | 1.38 (1.15 to 1.65) | <0.0001 | 1.38 (1.15 to 1.65) | <0.0001 |

For a continuous or an ordinal variable, the OR is calculated relative to a one-unit change in the variable. For ACSCs, the reference category is absence of the condition recorded.
Goodness of fit: likelihood ratio test—LR $\chi^2$ 144.6, p <0.0001.
*Proportional odds assumption violated for age and dementia diagnostic group. Note that where it is not violated, the same OR applies to both comparisons.
ACSC, ambulatory care sensitive condition; UTI, urinary tract infection.

## Sensitivity analysis (expanded ACSC definition)

In addition to perforated/bleeding ulcer (OR 6.92), pneumonia (OR 1.38) and UTI pyelonephritis (OR 1.49), in the sensitivity analysis where the ACSC definition was relaxed, convulsion/epilepsy (OR 1.58) was also found to be significantly associated with more severe dementia (online supplemental table 2).

## DISCUSSION

We evaluated all-cause and potentially avoidable causes of hospitalisation around the times of randomly selected cognitive function assessments in people with clinically diagnosed dementia, modelling mild versus moderate versus severe dementia at the index dates in question. Over half of our sample had a hospitalisation of any kind during a 12-month period of evaluation, and almost a quarter had an ACSC-related hospital admission, which accounted for nearly 40% of hospitalised patients. The proportions of patients with at least one hospitalisation overall, or with one ACSC-related hospitalisation, were greater in the groups with more advanced dementia. The odds of being in a more advanced stage was almost 1.3-fold higher for those with an ACSC-related admission after adjusting for demographics, time from first dementia record at index date (a proxy for disease duration) and neighbourhood deprivation.

Around 6% of community-dwelling individuals with dementia in a US Veterans Health Administration database had at least one ACSC admission during a 12-month period; however, Medicare facility use was not included, and hospitalisations may, thus, have been underestimated.[23] In another study, an average of 28% of Medicare beneficiaries per year aged ≥65 years with dementia participating in the Washington Heights-Inwood Columbia Aging Project had an ACSC-related admission,[8] which was more consistent with our own findings.

After adjusting for a range of clinical and demographic variables, we found that an ACSC-related admission remained associated with more advanced dementia. Hospitalisations with UTI/pyelonephritis, pneumonia and perforated/bleeding ulcer recorded as part of an admission were associated with more advanced dementia in our study. Although these have been reported as common causes of admission in people with dementia relative to those without dementia,[24–26] there is sparse literature on relationships with dementia severity. A study of Medicare beneficiaries with dementia found that worse cognitive status was associated with higher risks of hospitalisations for diabetes, pneumonia, hypertension and dehydration, and that worse functional status was associated with higher risks of hospitalisation for diabetes and UTI, marginally higher for dehydration, but lower for hypertension.[8]

UTI/pyelonephritis was recorded in 17.4% of hospitalised patients with dementia in our sample, and its presence was associated with more severe dementia. UTIs in more severe dementia may reflect worse immune function,[27] problems with mobility associated with urinary incontinence and subsequent infection[28] and inactivity or dehydration.[29] Consistently, we also observed a suggestion of association with dehydration in more severe dementia, although this was not significant (p=0.09). The presence of acute urinary symptoms and laboratory evidence (such as a positive urine culture) is required to make a formal UTI diagnosis. Difficulties collecting a urine specimen, an inability to accurately report symptoms, together with the high prevalence of asymptomatic bacteriuria in people with dementia, may lead to misdiagnosis, particularly in those whose dementia is severe.[30 31] Furthermore, non-specific symptoms such as new onset or worsening of confusion is a common reason suspecting a UTI in elderly patients and may also lead to overdiagnosis in this population.[32] These factors may confound our findings and overestimate the impact of UTIs on ACSC-related admissions.

Hospitalised pneumonia was common in our sample (12.5% of hospitalised patients with dementia) and was associated with more advanced dementia, as has been described by others.[33 34] Uptake of the pneumococcal vaccine in the over-65 age-group is relatively high in the UK (69.5% of the primary care population)[35]: this suggests that pneumonia in our sample may be due to other bacterial or viral causes or may be occurring as a result of an impaired immune system, dysphagia or urinary incontinence, which are common in, and associated with a higher risk of pneumonia in, the elderly.[36 37] Furthermore, it was not possible to determine whether instances of pneumonia identified in our sample were acquired within hospital as part of the inpatient episode or in the community prior to hospitalisation.

Perforated/bleeding ulcer-related admissions were strongly associated with more severe dementia at the index date. A lack of literature on this topic meant that we were unable to corroborate our findings. Importantly, as only six individuals had an admission for this ACSC, limited conclusions can be drawn from our analysis. An understanding of the use of non-steroidal anti-inflammatory drugs (NSAIDs) (as a risk factor for GI bleeding) and use of and adherence to proton pump inhibitors which are indicated for the management and/or prevention of GI bleeding by dementia severity would be useful to provide context to our observations.

Lastly, we observed that the mean number of ACSC admissions among hospitalised patients was highest in those with moderate compared with mild or severe dementia. It is possible that this observation reflects the increased support that people with more advanced dementia receive.[38] For example, compared with those with moderate dementia, individuals with severe dementia may quality for support (such as home healthcare) or be more likely to live in assisted accommodation or care homes where there is greater support to manage ACSCs on a day-to-day basis. For those residing in the community, caregiver burden has been reported to be associated with an increased risk of hospitalisation.[39 40] Barriers accessing care (eg, individuals being too frail to

be transported to hospital) and end-of-life care pathway implementation, such as individual care goals, may also influence hospitalisation patterns.

There is an existing body of evidence evaluating various models of care on the risk of hospitalisation including care management, counselling/self-help, enhanced general practitioner (GP) services or memory clinics and physiotherapy or occupational therapy, although none has demonstrated a clear benefit.[41] GP practice factors such as a larger practice population (which may have greater experience with dementia care and/or share patient care between GPs), quality of care and/or an increasing proportion of the local clinical budget allotted to mental health are reported to be associated with a slight decrease in the rate of avoidable and unavoidable hospital admissions.[42] Others have reported poor continuity of care as a risk factor for hospitalisation.[43] Future care models could include a multiagency approach including caregiver support.

## Strengths and limitations

Our study evaluated hospitalisations in a large sample of clinically diagnosed cases of dementia where cognitive assessment is performed as part of routine clinical care. Linkage of our clinical data set to a national hospitalisation database benefited from near-complete outcome ascertainment, since hospitalisation is free at the point of delivery in the UK context. We also benefited from information allowing differentiation between mild, moderate and severe dementia. Considering limitations, the analysed sample was from a single urban and suburban catchment and included only cases with dementia diagnosed in specialist services; however, the estimated proportion of people with dementia in the SLaM catchment who receive a specialist diagnosis is relatively high at 75.2%.[44] Considering reasons for hospitalisation, these were derived from the discharge diagnoses recorded for last episode of the hospital admission, and it was not possible to distinguish complications that arose during the hospital stay from reasons for the initial admission. Furthermore, scrutiny of a patient's medical history prior to hospitalisation would be required to determine whether a particular ACSC-related admission was truly preventable. Thus, ACSC-related admissions serve as indicators of, rather than definitive measures of, suboptimal care in the primary care or community care settings. We used an established definition of ACSC-related admissions to aid comparability with other research. However, the relevance of some of these conditions for an older population is not clear. It is arguable that more specific ACSC definitions might be needed in this population.

Considering residual confounding, this analysis did not account for comorbidity burden or medication use (including polypharmacy), which are potentially associated with hospitalisations and dementia severity,[9 11 23 40 45] nor did it account for prior healthcare utilisation, functional ability[45] or psychiatric symptoms. Presence of frailty, living arrangement, available support (family, community and home health support), care giver burden and GP practice-level factors are associated with hospitalisation[40 42] but were not available in this analysis.

Patients with severe dementia were under-represented in our sample and the time since first diagnosis in SLaM in these patients was relatively short despite the extent of cognitive impairment; it is, therefore, possible that our patients with severe dementia are generally those presenting late in their disease course, and not representative of the general advanced dementia population. Our analysis evaluated associations between hospitalisation patterns and contemporaneously measured severity of dementia, so it is not appropriate to draw conclusions about causation. In addition, we evaluated hospitalisation outcomes up to 6 months after a cognitive measure and did not attempt to account for survival effects. Mortality increases with severity of dementia[34] and is a competing risk for hospitalisation, although the relatively short timeframe evaluated should mitigate the effects of this.

## CONCLUSIONS

Our study adds to a small but increasing body of evidence on the relationship between dementia and potentially avoidable hospitalisations. Despite the clinical and economic burden, we are not aware of any studies in the UK setting that describes potentially avoidable hospitalisations in patients with dementia. We show that potentially avoidable hospitalisations are common in people with dementia, and, in particular, in those with more advanced dementia; these findings are of importance given that hospitalisation is a significant element in the cost of dementia care and that the number of people with dementia is increasing. Greater attention to the extent of cognitive impairment, and not only the presence of a dementia diagnosis, is needed in order to prevent hospitalisations. However, more work is clearly needed to better understand factors contributing to the high rates of ACSC-related hospitalisation, including the management of ACSCs in the primary care setting. Reducing admissions for ACSCs represents an opportunity to improve both the quality and efficiency of care in healthcare systems.

**Acknowledgements** The authors would like to acknowledge Hitesh Shetty for his contributions to the study.

**Contributors** UG acted as the guarantor and was responsible for the design, analysis, interpretation and drafting of the manuscript. GP contributed to the drafting of the manuscript. NWG provided statistical input and contributed to the drafting of the manuscript. RS contributed to the design, interpretation and drafting of the manuscript.

**Funding** This work was supported by NIHR Biomedical Research Centre at the South London and Maudsley NHS Foundation Trust and King's College London. RS is additionally part-funded by an NIHR Senior Investigator Award, by the National Institute for Health Research (NIHR) Applied Research Collaboration South London (NIHR ARC South London) at King's College Hospital NHS Foundation Trust, and by the DATAMIND HDR UK Mental Health Data Hub (MRC grant MR/W014386).

**Disclaimer** The funder of the study had no role in the study design, data collection, data analysis, data interpretation, or writing of the report.

**Competing interests** UG and NWG are employees of GSK, hold stock and receive a salary from GSK. RS has received research support in the last 5 years from Janssen, Takeda and GSK.

**Patient and public involvement** Patients and/or the public were not involved in the design, or conduct, or reporting, or dissemination plans of this research.

**Patient consent for publication** Not applicable.

**Ethics approval** The Oxford Research Ethics Committee C (reference 18/SC/0372) approved CRIS at the Maudsley as a data resource for secondary analysis.

**Provenance and peer review** Not commissioned; externally peer reviewed.

**Data availability statement** Data may be obtained from a third party and are not publicly available. Because of their nature, and to comply with their ethical approval, CRIS data are required to remain within the firewall of the South London and Maudsley NHS Foundation Trust (SLaM). Access to the data used for this study can be facilitated by the CRIS Oversight Committee on application and with appropriate SLaM affiliation, details of which can be obtained from cris. administrator@slam.nhs.uk.

**ORCID iDs**
Usha Gungabissoon http://orcid.org/0000-0002-2040-1763
Gayan Perera http://orcid.org/0000-0002-3414-303X

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
