## [Reviewer comments · BMJ Open]

ARTICLE DETAILS

TITLE (PROVISIONAL)	POTENTIALLY AVOIDABLE CAUSES OF HOSPITALISATION IN PEOPLE WITH DEMENTIA: CONTEMPORANEOUS ASSOCIATIONS BY STAGE OF DEMENTIA IN A SOUTH LONDON CLINICAL COHORT
AUTHORS	Gungabissoon, Usha; Perera, Gayan; Galwey, Nicholas; Stewart, Robert

VERSION 1 – REVIEW

REVIEWER	Hullick, Carolyn University of Newcastle Medical Society
REVIEW RETURNED	16-Sep-2021

GENERAL COMMENTS	Thanks for the opportunity to review this manuscript. Hospitalisation of people with dementia is an important area where there is limited evidence. Abstract: Thanks for reviewing the conclusion. I think that in people with dementia, presuming that a hospital admission is unnecessary or avoidable needs careful consideration. Please review the wording regarding this. Page 6: strengths and limitations: I would use the word episode rather than spell in line 27 and throughout the document. Perhaps spell is appropriate language for the UK. The editor can provide advice on this. Line 39 and throughout, I think there needs to be further expansion on institutional residence. For people with dementia, I would mention care homes (or residential homes and nursing homes) as this is likely to be the commonest place other than home that people with dementia live. You could say institutional residence, such as care homes. Introduction: Page 7, line 39. This paragraph needs a sentence about the context of an ACSC for people with dementia. People with dementia with ACSC, hospitalisation may be not avoidable due to other factors like their frailty, care stress. I agree with the next paragraph. The challenge is that even if they need admission, they are at greater risk of hospital acquired complications than older people without dementia. They are at particularly high risk of delirium.
--

	Page 8, line 27. The other factors to consider are  1. Clinicians concerns regarding risk of discharge 2. Family concerns regarding risk of discharge as well as carer stress. Methods: Data source: I could not see any mention of death. This patient population also has a high mortality. Were they excluded from the dataset. I am presuming so but could not see that it was mentioned. Please explain how death was managed in this dataset. Please clarify for the reader (me!). I think SLaM is the mental healthcare provider. It has the database with patients with dementia. This dataset was linked to the acute hospital dataset. I presume that the acute hospital geography is not described as this could be much broader than SLaM. Is that correct? Because the SLaM coding was incomplete, you also needed to use NLP to identify a more complete list of people with dementia. Is that correct? How did you validate this dataset? Once you had the SLaM dataset, then CRIS was used for the hospital data and this dataset is complete. Is that correct? Page 12, line 7, In addition, patients who were in active contact with acute hospital liaison services at the time of the initial diagnosis were excluded. Is that the dementia diagnosis, not the ACSC diagnosis. Why were they excluded? Please explain how a hospital diagnosis of dementia is different to a community one? Page 13, line 20. Please justify the MMSE scores that you have used. These are not the cut offs that I would use clinically. I would also be interested to know how many of the patients had dementia codes in their hospital ICD coding. One of the problems for people with dementia is that it is not recognised and sometimes not recorded in hospital therefore developing models of care for ACSC can be difficult. Results: As mentioned above, I would like to see death in table 1 but it may be that these patients have been excluded. Table 2, I would take flu and gangrene out of the table. I can not imaging that gangrene admission would be avoidable. Discussion: I agree with your assessment in the results. Line 20, page 28, I don't think you can comment on bleeding ulcer given the small number of patients in the dataset. Line 51, page 28. An important point to consider around the diagnosis of UTI is that UTIs tend to be overdiagnosed in older people, particularly those with dementia. Instead they have bacteriuria. Given the challenges of communication, particularly with severe dementia, UTIs are likely to be overdiagnosed, in
--	--

	someways, to give the patient a diagnosis for hospital admission. Some discussion regarding this would be helpful. Page 30, line 10, as above, I am not sure that you can draw much conclusion about bleeding ulcers. If you are going to talk about medication, I would talk about polypharmacy and the risk of medication interactions. I think the limitation about care home needs a little more discussion. For me, its important work. I think you could elaborate a little more on alternative models of care. One of the challenges for me about this group is that perhaps the coding is not good enough to pick up the reasons for hospitalisation like increasing behavioural disturbance, increasing carer stress and the lack of recognition of the diagnosis for dementia when people are in acute hospitals. Well done, and thank you for your important work
--	---

REVIEWER	Arendts, Glenn The University of Western Australia, Medical School
REVIEW RETURNED	10-Jan-2022

GENERAL COMMENTS	Overall I found this an extremely well written manuscript and have very few criticisms. The authors have done a good job in acknowledging the limitations of their data, particularly the absence of other critical variables such as comorbidity which would strongly influence hospitalisation for ACSC. Although the authors pre-empt this criticism by explaining their approach in the methods section, I still find the presentation of the main results is quite atypical, brought about by modelling dementia stage as the primary outcome. It does disorient the reader to read in the abstract, for example, that "an ACSC-related admission was associated with 1.3-fold increased odds of more severe dementia after adjusting for demographic factor" when the objective is "to investigate differences {in ACSC hospitalisations} by stage of dementia". The objective in other words reflects the clinical relevance of these data : is more severe dementia associated with more avoidable hospitalisation? Where I think the manuscript could be improved is in the discussion. Firstly, UTI is a vexatious diagnosis in older adults with dementia. The gold standard of UTI diagnosis is rarely applied because of difficulties in correctly collecting a specimen for culture, the high prevalence of asymptomatic bacteriuria in this population, and the inability to accurately report symptoms when dementia is severe (see for example PMID 31478344). As there is no verification standard for a hospital discharge ICD10 code, at least some of these hospitalisations are likely to be misdiagnosis, and this is more likely to be in the severe dementia cohort, confounding the findings. Secondly, the most interesting finding in the study is that "The mean number of ACSC admissions amongst hospitalised patients was highest in those with moderate compared to mild or severe dementia". This replicates findings in other studies and is likely because many ACSC hospitalisations (as opposed to managing the condition as an outpatient) in this group are at least partly
--

	driven by care stress and lack of supports. The severe group being more likely to have more social supports including living in an aged care facility that may enable outpatient care, whereas the moderate group are not qualifying for the same level of support and so carer burden (or even the absence of a carer) is incorporated into the decision to hospitalise. This could be addressed in the discussion, and the absence of living arrangement/social support data to include in your model be acknowledged as another limitation.
--	---

VERSION 1 – AUTHOR RESPONSE

Reviewer: 1

Dr. Carolyn Hullick, University of Newcastle Medical Society

Comments to the Author: Thanks for the opportunity to review this manuscript. Hospitalisation of people with dementia is an important area where there is limited evidence.

Abstract:

- Thanks for reviewing the conclusion. I think that in people with dementia, presuming that a hospital admission is unnecessary or avoidable needs careful consideration. Please review the wording regarding this.

- Response: Thank you for flagging this. We have revised the wording accordingly.

Page 6: strengths and limitations:

- I would use the word episode rather than spell in line 27 and throughout the document. Perhaps spell is appropriate language for the UK. The editor can provide advice on this.

- Response: We have replaced hospital spell with admission as this may be a more widely used term in settings outside of the UK.

- Line 39 and throughout, I think there needs to be further expansion on institutional residence. For people with dementia, I would mention care homes (or residential homes and nursing homes) as this is likely to be the commonest place other than home that people with dementia live. You could say institutional residence, such as care homes.

- Response: We thank you for your suggestion. We've incorporated your suggested wording in the strengths and limitations section and in the main body of the manuscript

Introduction:

- Page 7, line 39. This paragraph needs a sentence about the context of an ACSC for people with dementia. People with dementia with ACSC, hospitalisation may be not avoidable due to other factors like their frailty, care stress. I agree with the next paragraph. The challenge is that even if they need admission, they are at greater risk of hospital acquired complications than older people without dementia. They are at particularly high risk of delirium.

- Response: We agree. In light of the reviewer's comments, we have further elaborated on this within the discussion.

- Page 8, line 27. The other factors to consider are

1. Clinicians concerns regarding risk of discharge

2. Family concerns regarding risk of discharge as well as carer stress.

- Response: We agree with the reviewer's comments that this may be a particular issue for those with more severe dementia.

Methods:

Data source:

- I could not see any mention of death. This patient population also has a high mortality. Were they excluded from the dataset. I am presuming so but could not see that it was mentioned. Please explain how death was managed in this dataset.

- Response: These individuals were not excluded given the relatively short period of observation. The following limitation is acknowledged in the discussion:

In addition, we evaluated hospitalisation outcomes up to 6-months after a cognitive measure and did not attempt to account for survival effects. Mortality increases with severity of dementia, and is a competing risk for hospitalisation, although the relatively short timeframe evaluated should mitigate the effects of this.

- Please clarify for the reader (me!). I think SLaM is the mental healthcare provider. It has the database with patients with dementia. This dataset was linked to the acute hospital dataset. I presume that the acute hospital geography is not described as this could be much broader than SLaM. Is that correct? Because the SLaM coding was incomplete, you also needed to use NLP to identify a more complete list of people with dementia. Is that correct? How did you validate this dataset?

- Response (setting and data source): We apologise and agree that the original wording was not very clear. We have revised the wording describing the setting (South London and Maudsley NHS Trust, SLaM) and the clinical database from which the data were abstracted from (CRIS). CRIS contains clinical data relating to mental healthcare provided within SLaM. We included individuals who received a dementia diagnosis within SLaM in our analysis.

- Response (NLP): We only used NLP to abstract measures of cognitive impairment; we have corrected the wording on the attached. It now reads:

Routine diagnoses recorded in SLaM are structured according to the International Classification of Diseases 10th edition (ICD-10) coding system, and are supplemented in CRIS by a natural language processing (NLP) to abstract measures of cognitive impairment.

- Once you had the SLaM dataset, then CRIS was used for the hospital data and this dataset is complete. Is that correct?

- Response: To obtain information on acute hospital admissions, we used a data-linkage between the CRIS and hospital episode statistics (HES) databases. HES contains all admissions in England, thus would capture whether a SLaM resident was hospitalised in a broader geography than SLaM. For example, hospitalisations in areas outside of the SLaM NHS Trust would be captured in our analysis.

- Page 12, line 7, In addition, patients who were in active contact with acute hospital liaison services at the time of the initial diagnosis were excluded. Is that the dementia diagnosis, not the ACSC diagnosis. Why were they excluded? Please explain how a hospital diagnosis of dementia is different to a community one?

- Response: We excluded patients who received their first dementia diagnosis from an acute hospital liaison service because referral to that service would have been predicated on a general hospital admission – i.e., would by definition be a group with significant hospitalised physical disorders and therefore potentially biasing any associations between dementia and comorbidity, even though the randomly selected index cognitive function scores might have reflected future, non-hospitalised time points. We apologise that this wasn't sufficiently clear and have included the following revised text in this respect:

In addition, we excluded patients who received their first dementia diagnosis from an acute hospital liaison service; these reflected cases where the recording of a dementia diagnosis might have been precipitated by a hospitalisation and by definition be a group with significant hospitalised physical disorders, therefore potentially biasing any associations between dementia and comorbidity.

- Page 13, line 20. Please justify the MMSE scores that you have used. These are not the cut offs that I would use clinically.

- Response: We acknowledge that there is heterogeneity in MMSE cut-offs used in clinical practice. Our understanding that an MMSE score of <10 is generally accepted as indicating severe cognitive impairment and a score of 10-20, moderate cognitive impairment, 1 2 however, there appears to be less agreement regarding accepted cut-offs for mild dementia and mild cognitive impairment (MCI).³ Typically, in the literature, scores of 21 to 26/27 have been used to define mild dementia and 27/28 to 30 for MCI to normal cognitive function. 2 4-6 We used cut-offs similar to those used by others 7, defining MMSE scores of 21-27, 10-20, and <10 for mild, moderate and severe dementia. We believe that the cut-offs that we adopted, although not used ubiquitously in clinical practice are clinically relevant and achieve the purpose of broadly classifying patients into mild, moderate and severe dementia stages.

- I would also be interested to know how many of the patients had dementia codes in their hospital ICD coding. One of the problems for people with dementia is that it is not recognised and sometimes not recorded in hospital therefore developing models of care for ACSC can be difficult.

- Response: Although this was not included in our analysis, our colleagues previously have evaluated this. Sommerlad et al reported that individuals with known dementia but with milder levels of cognitive impairment were either less likely to have dementia recorded as part of a hospital admission, and/or less likely to have dementia recorded in the primary or secondary position as compared to individuals displaying agitation or aggression, or with higher levels of cognitive impairment or those of a more advanced age.⁸

Results:

- As mentioned above, I would like to see death in table 1 but it may be that these patients have been excluded.

- Response: this has been addressed in an earlier comment.

- Table 2, I would take flu and gangrene out of the table. I can not imaging that gangrene admission would be avoidable.

- Response: As recommended, we have taken these out of the table, and instead added this as a footnote.

Discussion:

- I agree with your assessment in the results. Line 20, page 28, I don't think you can comment on bleeding ulcer given the small number of patients in the dataset.

- Response: We thank the reviewer for their comment. We have revised the wording in the existing paragraph to emphasise this point.

- Line 51, page 28. An important point to consider around the diagnosis of UTI is that UTIs tend to be overdiagnosed in older people, particularly those with dementia. Instead they have bacteriuria. Given the challenges of communication, particularly with severe dementia, UTIs are likely to be overdiagnosed, in someways, to give the patient a diagnosis for hospital admission. Some discussion regarding this would be helpful.

- Response: We are grateful for the reviewer's feedback. We have included a paragraph describing the challenges of a UTI diagnosis and the potential for over-diagnosis in this population, together with the relevance of this to our findings.

- Page 30, line 10, as above, I am not sure that you can draw much conclusion about bleeding ulcers. If you are going to talk about medication, I would talk about polypharmacy and the risk of medication interactions.

- Response: This has been addressed in an earlier comment.

- I think the limitation about care home needs a little more discussion.
- Response: We thank the reviewer for this suggestion. We have expanded on this in the discussion.

- For me, its important work. I think you could elaborate a little more on alternative models of care. One of the challenges for me about this group is that perhaps the coding is not good enough to pick up the reasons for hospitalisation like increasing behavioural disturbance, increasing carer stress and the lack of recognition of the diagnosis for dementia when people are in acute hospitals.
- Response: We grateful for this feedback. We have sought to emphasise a number of the aspects raised by the reviewer throughout the discussion including caregiver stress and challenges relating to use of structured/coded data. We have also included an additional paragraph relating to alternative models of care.

- Well done, and thank you for your important work
- Response: We thank the reviewer for their comments and agree that this is an important topic where there is limited evidence.

Reviewer: 2

Dr. Glenn Arendts, Western Australian Institute for Medical Research, University of Sydney

- Comments to the Author: Overall I found this an extremely well written manuscript and have very few criticisms. The authors have done a good job in acknowledging the limitations of their data, particularly the absence of other critical variables such as comorbidity which would strongly influence hospitalisation for ACSC.

- Response: We thank the reviewer for their kind comments.

- Although the authors pre-empt this criticism by explaining their approach in the methods section, I still find the presentation of the main results is quite atypical, brought about by modelling dementia stage as the primary outcome. It does disorient the reader to read in the abstract, for example, that "an ACSC-related admission was associated with 1.3-fold increased odds of more severe dementia after adjusting for demographic factor" when the objective is "to investigate differences [in ACSC hospitalisations] by stage of dementia". The objective in other words reflects the clinical relevance of these data : is more severe dementia associated with more avoidable hospitalisation?

- Response: To address this concern (which the reviewer correctly alluded to) we have added brief context in the abstract to help orientate the reader.

- Where I think the manuscript could be improved is in the discussion.

Firstly, UTI is a vexatious diagnosis in older adults with dementia. The gold standard of UTI diagnosis is rarely applied because of difficulties in correctly collecting a specimen for culture, the high prevalence of asymptomatic bacteriuria in this population, and the inability to accurately report symptoms when dementia is severe (see for example PMID 31478344). As there is no verification standard for a hospital discharge ICD10 code, at least some of these hospitalisations are likely to be misdiagnosis, and this is more likely to be in the severe dementia cohort, confounding the findings.

- Response: We are grateful for the reviewer's feedback. We have included a paragraph describing the challenges of a UTI diagnosis and the potential for over-diagnosis in this population, together with the relevance of this to our findings. We have also included the suggested reference.

- Secondly, the most interesting finding in the study is that "The mean number of ACSC admissions amongst hospitalised patients was highest in those with moderate compared to mild or severe dementia". This replicates findings in other studies and is likely because many ACSC hospitalisations (as opposed to managing the condition as an outpatient) in this group are at least partly driven by care stress and lack of supports. The severe group being more likely to have more social supports including living in an aged care facility that may enable outpatient care, whereas the moderate

group are not qualifying for the same level of support and so carer burden (or even the absence of a carer) is incorporated into the decision to hospitalise. This could be addressed in the discussion, and the absence of living arrangement/social support data to include in your model be acknowledged as another limitation.

- Response: We are grateful for the reviewer's insights and suggestions. We have added wording in the discussion to specifically address these points.

References

1. NICE-SCIE. A NICE–SCIE Guideline on supporting people with dementia and their carers in health and social care. National Clinical Practice Guideline, 2015.
2. Prince MK, M.; Guerchet, M.; McCrone, P.; Prina, M.; Comas-Herrera, A.; Wittenberg, R.; Adelaja, B.; Hu, B.; King, D.; Rehill, A.; Salimkumar, D.; Dementia UK Update, 2014.
3. Creavin SW, S.; Noel-Storr, AH.; Trevelyan, CM.; Hampton, T.; Rayment, D.; Thom, VM.; Nash, KJ E.; Elhamoui, H.; Milligan, R.; Patel, AS; Tsivos, DV.; Wing, T.; Phillips, E.; Kellman, SM.; Shackleton, HL.; Singleton, GF.; Neale, BE.; Watton, ME.; Cullum S.; Mini-Mental State Examination (MMSE) for the detection of dementia in clinically unevaluated people aged 65 and over in community and primary care populations. . Cochrane Database of Systematic Reviews 2016(1):Issue 1. doi: DOI: 10.1002/14651858.CD011145.pub2
4. Arevalo-Rodriguez I, Smailagic N, Roque IFM, et al. Mini-Mental State Examination (MMSE) for the detection of Alzheimer's disease and other dementias in people with mild cognitive impairment (MCI). Cochrane Database Syst Rev 2015(3):CD010783. doi: 10.1002/14651858.CD010783.pub2 [published Online First: 2015/03/06]
5. Vertesi A, Lever JA, Molloy DW, et al. Standardized Mini-Mental State Examination. Use and interpretation. Can Fam Physician 2001;47:2018-23. [published Online First: 2001/11/29]
6. NICE. Donepezil, galantamine, rivastigmine and memantine for the treatment of dementia. Health technology appraisal 2018 update. In: NICE, ed. 2018 ed, 2018.
7. NICE. Donepezil, galantamine, rivastigmine and memantine for the treatment of dementia. Health technology appraisal 2018 update, 2018.
8. Sommerlad A, Perera G, Singh-Manoux A, et al. Accuracy of general hospital dementia diagnoses in England: Sensitivity, specificity, and predictors of diagnostic accuracy 2008-2016. Alzheimers Dement 2018;14(7):933-43. doi: 10.1016/j.jalz.2018.02.012 [published Online First: 2018/04/29]

VERSION 2 – REVIEW

REVIEWER	Hullick, Carolyn University of Newcastle Medical Society
REVIEW RETURNED	08-Feb-2022

GENERAL COMMENTS	Thanks for the opportunity to review this manuscript on hospitalisation for people with dementia, particularly comparing those at different stages of dementia. Health service planning for hospitalisation for people with dementia is important given the expectation that there will be increasing numbers of people with cognitive impairment and the risks and benefits of hospital care and the importance in maintaining equity of access to care as people age. I enjoyed reading the paper and thinking about what the messages are that the findings implied. I have a few comments that I think will strengthen the paper. Data source (page 6) It was not clear to me how the CRIS data for SLaM. I understand the hospitalisation data . I was not sure how people came to be on
--

	the CRIS database. Was this outpatient visits, general practice visits, specialist visits? Are patients referred to this service or is it a primary care dataset? I understand that they became eligible once they had a MMSE or HoNOS undertaken and that the scores needed to reflect dementia. Thanks for including an extra sentence on the population of patients in CRIS. Measurements (Page 8) MMSE: It says greater than 20 but I believe that it is MMSE 21 to 27. Thanks for clarifying. Acute general inpatient admissions: Is this an overnight admission or any hospital admission including day-only. Were things like dialysis and chemotherapy included in this data set? All cause and ACSC-related hospitalisation. Page 13 I believe that 57% of patients had any hospital admission. Is this correct? Page 16: Moderate versus severe dementia is not analysed in the association between acsc hospitalisation and dementia stage. I think that it is clinically important to identify that people with severe dementia had less admissions than those with moderate dementia. Perhaps the discussion section could mention the challenges that people with severe dementia have in accessing care when they are often too frail to be transported to hospital and access specialist care that cannot come to them. This is also related to the challenges that people with severe have in accessing clinical care. Discussion (page 20) As the authors imply, some of the diagnosis of UTI in people with dementia may be misdiagnosis or overdiagnosis. The importance of this is that it will overestimate the impact of UTIs on ACSC. Page 21: I think that it is important that the highest rate of hospitalisation was in the moderate group. As mentioned above, some is also related to reduced equity of access to care for people with severe dementia, in some ways reflected in the complex logistics and the significant disturbance transport from home can incur. Convulsions: Convulsions was be a clinical complication of more severe dementia which are more difficult to manage that other forms of convulsion. I also wondered whether there needs to be a sentence on advance care planning and identifying a person's goals of care which may not be reflected in hospitalisation. As the authors mention in the introduction, ACSC is about acute management of conditions as well as chronic disease management. These diagnoses may not be diagnoses that are as relevant for someone with dementia as they are in a younger person for example pressure injuries and constipation would be conditions that are potentially avoidable in this patient population.
--	---

REVIEWER	Arendts, Glenn The University of Western Australia, Medical School
REVIEW RETURNED	26-Jan-2022

GENERAL COMMENTS	The revisions made have improved the manuscript and in my opinion render it appropriate for publication
---

VERSION 2 – AUTHOR RESPONSE

Reviewer: 1

- Thanks for the opportunity to review this manuscript on hospitalisation for people with dementia, particularly comparing those at different stages of dementia. Health service planning for hospitalisation for people with dementia is important given the expectation that there will be increasing numbers of people with cognitive impairment and the risks and benefits of hospital care and the importance in maintaining equity of access to care as people age. I enjoyed reading the paper and thinking about what the messages are that the findings implied. I have a few comments that I think will strengthen the paper.

- Response: We thank the reviewer for their invaluable feedback and suggestions to strengthen our paper.

- Data source (page 6) It was not clear to me how the CRIS data for SLaM. I understand the hospitalisation data . I was not sure how people came to be on the CRIS database. Was this outpatient visits, general practice visits, specialist visits? Are patients referred to this service or is it a primary care dataset? I understand that they became eligible once they had a MMSE or HoNOS undertaken and that the scores needed to reflect dementia. Thanks for including an extra sentence on the population of patients in CRIS.

- Response: We thank the reviewer for this feedback. As suggested, we have included additional wording describing the CRIS database and the patients whose data are included.

The study sample comprised of residents within the South London and Maudsley NHS Foundation Trust (SLaM) who had received a dementia diagnosis in specialist services. The trust is a provider of mental healthcare, dementia assessment and management, for a south London catchment of >1.2 million residents in Lambeth, Lewisham, Croydon and Southwark; EHRs have been implemented across all SLaM services since 2006. The Clinical Record Interactive Search (CRIS) data resource, used to identify dementia cases, provides research access to anonymised electronic health records (EHRs); it allows both structured and unstructured data to be abstracted from patient records based on interactions within secondary care mental health services within SLaM.

- Measurements (Page 8) MMSE: It says greater than 20 but I believe that it is MMSE 21 to 27. Thanks for clarifying.

- Response: The reviewer is correct, that this relates to MMSE 21-27. We have revised the text accordingly.

- Acute general inpatient admissions: Is this an overnight admission or any hospital admission including day-only. Were things like dialysis and chemotherapy included in this data set?

- Response: The HES inpatient dataset included all hospital admissions (elective and non-elective) regardless of duration. Although admissions for dialysis would be captured in this dataset, they would not have been considered to be ACSC-related.

• All cause and ACSC-related hospitalisation. Page 13 I believe that 57% of patients had any hospital admission. Is this correct?

- Response: That is correct, amongst the overall sample of patients with dementia included in our analysis, 57% had a hospital admission (of any cause). We have provided additional clarity in the revised manuscript within the results section.

• Page 16: Moderate versus severe dementia is not analysed in the association between acsc hospitalisation and dementia stage. I think that it is clinically important to identify that people with severe dementia had less admissions than those with moderate dementia. Perhaps the discussion section could mention the challenges that people with severe dementia have in accessing care when they are often too frail to be transported to hospital and access specialist care that cannot come to them. This is also related to the challenges that people with severe have in accessing clinical care.

- Response: In the discussion, we consider whether the observed differences may reflect the fact that people with more severe dementia are more likely to receive home health care or live in assisted accommodation and thus have more support to manage comorbidities than an individual living alone in a relatively socially unsupported urban environment. However, as highlighted by the reviewer, this could also be explained by lack of access to emergency care or end-of-life care pathway implementation. Unfortunately, it is not possible to distinguish between these in our analysis. We have now included additional wording describing barriers accessing care and end-of-life pathway implementation as additional considerations within the discussion.

• Discussion (page 20) As the authors imply, some of the diagnosis of UTI in people with dementia may be misdiagnosis or overdiagnosis. The importance of this is that it will overestimate the impact of UTIs on ACSC.

- Response: We have added the following wording to conclude the UTI paragraph:

These factors may confound our findings and overestimate the impact of UTIs on ACSC-related admissions

• Page 21: I think that it is important that the highest rate of hospitalisation was in the moderate group. As mentioned above, some is also related to reduced equity of access to care for people with severe dementia, in some ways reflected in the complex logistics and the significant disturbance transport from home can incur.

- Response: We agree that this could be a contributing factor, together with other potential reasons provided above. Future research to specifically explore this is needed.

• Convulsions: Convulsions was be a clinical complication of more severe dementia which are more difficult to manage than other forms of convulsion. I also wondered whether there needs to be a sentence on advance care planning and identifying a person's goals of care which may not be reflected in hospitalisation.

- Response: Thank you for highlighting this. As per our earlier response, we have now included wording relating to end-of-life care implementations and individual care goals in the discussion.

• As the authors mention in the introduction, ACSC is about acute management of conditions as well as chronic disease management. These diagnoses may not be diagnoses that are as relevant for someone with dementia as they are in a younger person for example pressure injuries and constipation would be conditions that are potentially avoidable in this patient population.

- Response: We agree with the reviewer's comment. We have included additional wording in the limitations section of the discussion.

We used an established definition of ACSC-related admissions to aid comparability with other research. However, the relevance of some of these conditions for an older population is not clear. It is arguable that more specific ACSC definitions might be needed for this population.

VERSION 3 – REVIEW

REVIEWER	Hullick, Carolyn University of Newcastle Medical Society
REVIEW RETURNED	14-Mar-2022

GENERAL COMMENTS	Thanks for the opportunity to review this next draft. I am happy with the changes that have been made. It is a lot clearer to follow
--